# Effects of Sex and Cross-Sex Hormone Treatment on Renal MCT/SMCT Expression Following Prepubertal Gonadectomy

**DOI:** 10.3390/pharmaceutics17020252

**Published:** 2025-02-14

**Authors:** Annie Lee, Qing Zhang, Hao Wei, Melanie A. Felmlee

**Affiliations:** Department of Pharmaceutical Sciences, Thomas J. Long School of Pharmacy, University of the Pacific, Stockton, CA 95211, USA; a_lee82@u.pacific.edu (A.L.); q_zhang6@u.pacific.edu (Q.Z.); h_wei1@u.pacific.edu (H.W.)

**Keywords:** monocarboxylate transporters, sex hormones, testosterone, 17β-estradiol, puberty

## Abstract

**Background:** Kidney proton- and sodium-dependent monocarboxylate transporters (MCT/SMCT) are involved in the renal reabsorption of substrates, and thus factors involved in their regulation may have pharmacokinetic implications. Previous studies have demonstrated sex hormone-dependent regulation of MCTs and SMCTs in tissues involved in drug disposition. The present study evaluates the impact of puberty on renal MCT/SMCT expression with ovariectomy and castration conducted before puberty, removing the initial exposure to sex hormones. **Methods:** Male and female rats were castrated or ovariectomized before puberty (4 weeks of age), and subsequently treated with testosterone, 17β-estradiol, progesterone, or both 17β-estradiol and progesterone for 21 days starting at 10 weeks of age. MCT1, CD147, and SMCT1 membrane-bound kidney expression were quantified by Western blot. **Results:** SMCT1 and CD147 expression were significantly higher in OVX and CST rats treated with testosterone, and testosterone plasma concentrations showed a significant positive correlation with MCT1, SMCT1, and CD147 expression. CD147 expression was significantly downregulated in OVX rats treated with estrogen, compared to placebo controls, and estrogen plasma concentrations were significantly negatively correlated with CD147 expression. **Conclusions:** Sex and cross-sex hormone treatment altered MCT1, CD147, and SMCT1 expression when gonadectomy was conducted before puberty. The magnitude and direction of the expression differences differed when compared to animals that underwent gonadectomy after puberty, suggesting that sex hormone exposure during puberty plays a key role in MCT1/SMCT1 renal expression. Further studies are needed to elucidate the underlying mechanisms for the differential regulation of MCTs/SMCTs when gonadectomy occurs before or after puberty.

## 1. Introduction

The solute carrier 16 (SLC16) family is comprised of 14 monocarboxylate transporters (MCT1-14, SLC16A1-14), and the solute carrier 5A (SLC5A) family includes sodium-dependent monocarboxylate transporters SMCT1 (SLC5A8) and SMCT2 (SLC5A12) [1,2]. MCTs depend on ancillary proteins for membrane expression [1]. CD147 (basigin) is an ancillary protein for MCTs 1, 3, 4, 11, and 12, and has been demonstrated to mediate the translocation of MCT1 and MCT4 for cell surface expression [1,3,4,5,6,7]. Embigin (gp70) is an ancillary protein for MCT2, and functions as an alternate chaperone for MCT1 when CD147 is not present [1,3]. MCTs 1-4 and SMCT1 transport pyruvate, lactate, short-chain fatty acids, and monocarboxylate drugs [1,2]. The other isoforms transport drugs such as probenecid and bumetanide (MCT6), ketone bodies (MCT 7), thyroid hormones T3 and T4 (MCT8), carnitine (MCT9), aromatic amino acids (MCT10), and creatinine (MCT12) [1].

MCT/SMCTs are expressed throughout the body, but isoform expression is tissue-specific [1]. At the proximal renal tubule, MCT1 and MCT4 are expressed on the basolateral membrane, while SMCT1 is localized to the apical membrane of the S2/S3 segments [8,9,10]. Kidney MCT/SMCTs mediate the renal reabsorption of exogenous drugs, including γ-hydroxybutyrate (GHB). Consequently, renal monocarboxylate transporter expression has pharmacokinetic implications, as the kidney is a major organ for drug excretion. GHB pharmacokinetic studies revealed variability in renal clearance between females in different stages of the estrous cycle, ovariectomized (OVX) females, intact males, and castrated (CST) males [11]. Proestrus females had significantly higher renal clearance than OVX females, CST males, intact males, and diestrus females following 600 mg/kg GHB i.v. [11]. These results indicated sex hormone mediated differences in GHB toxicokinetics and renal clearance, which are likely attributable to sex hormone-dependent regulation of kidney MCT/SMCT expression [11].

The effects of sex hormones on monocarboxylate transporter expression have been evaluated in hindlimb muscles [12], heart [12], and brain [13], but there is limited kidney expression data [14,15]. Kidney smct1 protein expression in orchiectomized mice was lower than in sham-operated mice, while testosterone-treated orchiectomized mice had greater expression compared to orchiectomized mice [14]. Alternatively, ovariectomized (OVX) mice treated with progesterone had significantly lower kidney smct1 protein expression compared to OVX mice [15]. We previously demonstrated sex hormone-mediated differences in MCT/SMCT protein expression in rats that underwent castration (CST) or OVX after the onset of puberty [16]. MCT1 expression in OVX rats was downregulated with estrogen and progesterone treatment and upregulated with testosterone treatment. CD147 expression was significantly greater in OVX rats treated with testosterone, compared to corresponding placebo. SMCT1 expression in CST rats was upregulated with estrogen, alone or in combination with progesterone, and downregulated with testosterone treatment. However, the role of the initial pubertal hormone exposure and the underlying regulatory mechanisms on the observed expression differences is unknown. Pubertal hormone exposure plays a vital role in the development of sex differences and has been demonstrated to impact epigenetic regulatory mechanisms, specifically, DNA methylation [17]. MCT1 regulation by DNA methylation has been demonstrated, and this may lead to differences in responses to exogenous sex hormones when gonadectomy is conducted before puberty.

The aim herein is to evaluate MCT/SMCT renal expression in response to sex and cross-sex hormone treatment in animals that have not gone through puberty, in order to evaluate the role of puberty in MCT/SMCT regulation. Renal membrane MCT1, CD147, and SMCT1 expression were evaluated in animal models that had undergone gonadectomy at a prepubescent age of 4 weeks [18,19] with subsequent hormone replacement with testosterone, 17β-estradiol, progesterone, or both 17β-estradiol and progesterone (or their respective placebo pellets).

## 2. Materials and Methods

### 2.1. Chemicals

LC/MS-grade acetonitrile was purchased from Fisher Scientific (Waltham, MA, USA). Testosterone-d3 and progesterone-d9 were obtained from Cerilliant (Round Rock, TX, USA). Ketamine, xylazine, and heparin were procured from Patterson Veterinary (Rocky Mountain, CO, USA).

### 2.2. Animals, Hormone Implantation, and Tissue Collection

OVX female and CST male Sprague–Dawley (SD) rats were procured from Charles River (Wilmington, MA, USA) at 5 weeks of age. Castration and ovariectomy surgeries were performed at 4 weeks of age at Charles River. Intact male and female Sprague–Dawley rats were procured from Charles River (Wilmington, MA, USA) at 8 weeks of age. Rats were housed in a temperature- and humidity-controlled environment with an alternating 12 h light/dark cycle and ad libitum access to rat chow and water. The Institutional Animal Care and Use Committee at the University of the Pacific approved all animal research conducted.

At 10 weeks of age, rats were subcutaneously implanted with a hormone pellet(s) in the region of the left shoulder blade, under isoflurane anesthesia. The pellets contained 1.5 mg 17β-estradiol and/or 50 mg progesterone, or 7.5 mg testosterone, or respective placebos formulated to release over 60 days (N = 6 per treatment group). The rats were exsanguinated and the tissue was collected under isoflurane anesthesia following 21 days of hormone exposure. Kidney samples were also collected from age-matched intact males and females in the estrus stage. Estrous cycles were monitored prior to tissue collection, using vaginal cytology methods described previously [11,20,21]. Kidneys were snap-frozen in liquid nitrogen and stored at −80 °C for subsequent experiments. Heparinized blood was centrifuged at 5000 rcf for 15 min at 4 °C to attain plasma; the samples were stored at −20 °C until further analysis.

### 2.3. Quantifications of Plasma Estrogen, Progesterone, and Testosterone

Plasma progesterone and testosterone concentrations were simultaneously measured using a previously validated LC-MS method, with minor modifications [16,22]. Briefly, standard and quality control (QC) stock solutions were prepared by diluting 200 µg/mL of the Steroids and Mixed Pharmaceutical Mix (Restek Corporation, Bellefonte, PA, USA) with acetonitrile. Standard and QC samples were prepared with 10 µL stock, 90 µL acetonitrile, 5 µL 100 ng/mL progesterone-d9, and 5 µL 100 ng/mL testosterone-d3. Plasma samples were prepared with 100 µL samples (or diluted with acetonitrile to be within the calibration range) of 5 µL 100 ng/mL progesterone-d9 and 5 µL 100 ng/mL testosterone-d3. Additionally, 1 mL of acetonitrile was added to each sample to precipitate proteins, and samples were centrifuged at 10,000 rpm for 20 min. In sum, 950 µL of supernatant was aspirated, dried under a stream of nitrogen, and reconstituted with 200 µL acetonitrile. The samples were analyzed with an Agilent 1200 series UPLC with an online degasser, binary pump, and autosampler coupled to an Agilent 6460 Triple Quad Mass Spectrometer. The chromatographic and MS conditions used were reported in a previous publication [16].

Plasma 17β-estradiol concentrations were measured with a competitive ELISA kit (Abcam, Cambridge, UK) in accordance with the manufacturer’s protocol. The concentrations were interpolated from the calibration curve using GraphPad Prism 10.3.1 (San Diego, CA, USA). Inter- and intra-day accuracy and precision based on quality control samples of the analyte were between 80 to 120%.

### 2.4. Western Blot

Kidney soluble and membrane protein fractions were extracted with the ProteoExtract Native Membrane Extraction kit (MilliporeSigma, Burlington, MA, USA). Total protein concentrations were determined with the Pierce BCA Protein Assay (Thermo Fisher Scientific, Waltham, MA, USA). Samples were diluted with deionized water and 4X Laemmli Sample buffer (BioRad, Hercules, CA, USA) containing 2-Mercaptoethanol (βME). Samples were heated at 37 °C for 30 min and 5 µg of protein was loaded into each lane for separation at 200 V on TGX Stain-Free FastCast Acrylamide 10% gels (BioRad, Hercules, CA, USA). Proteins were transferred to Odyssey Nitrocellulose Membranes (LI-COR, Lincoln, NE, USA) at 100 V for 25 min. Blots for SMCT1 analyses were cut above/at 100 MW and below 25 MW prior to blocking due to the presence of nonspecific binding. Membranes were blocked with 5% milk in 1X PBST for 1 or 1.5 h under rotation at room temperature prior to incubation with primary antibody for 15 to 16 h at 4 °C. The primary antibodies used for semi-quantification were anti-MCT1 (AB3540P, MilliporeSigma, Burlington, MA, USA), anti-CD147 (OAAJ01996, AVIVA, San Diego, CA, USA), and anti-SMCT1 (ARP44110_P050, AVIVA, San Diego, CA), and ATP1A1/ATP1A2/ATP1A3 (H-3) (sc-48345, Santa Cruz, Dallas, TX, USA) was used for validation; the antibodies were diluted with 1% milk in 1X PBST at 1:1350 for anti-MCT1 and at 1:1000 for the other antibodies. The membranes were washed with 1X PBST (3 to 4 times, 10 min each) before and after incubation with the secondary antibody. Membranes were incubated with secondary antibodies for 1 h at room temperature; goat anti-rabbit IgG H&L(HRP) (ab97051, abcam, Cambridge, UK) diluted at 1:10,000 was used for MCT1 analysis, and peroxidase AffiniPure goat anti-rabbit IgG (111-035-144, Jackson ImmunoResearch, West Grove, PA, USA) was diluted at 1:5000 for CD147 analysis and at 1:10,000 for SMCT1 analysis. The membranes were incubated with Clarity ECL reagent (BioRad, Hercules, CA, USA) for 2 min; SMCT1 blots were incubated with Radiance Plus Femtogram HRP substrate (Azure, Dublin, CA, USA) for 3 min. The chemiluminescent bands were imaged with a ChemidocTM Touch Imaging System (BioRad, Hercules, CA, USA). Band density and total lane density were determined with BioRad Image 6.0.1 software.

### 2.5. Data Analysis

Results are presented as mean ± standard deviation (SD) with individual data points. Individual protein band densities were normalized to total lane densities and adjusted using the inter-blot control to account for loading variability and variation across blots. Analyses were performed with GraphPad Prism 10.3.1 (San Diego, CA, USA). For each target protein and hormone treatment (testosterone, 17β-estradiol, progesterone, or 17β-estradiol and progesterone combined), mean density values of hormone-treated, hormone placebo-treated, and male and female controls were compared using one-way analysis of variance (ANOVA) with a Tukey post hoc test; *p*-values < 0.05 were considered significant. Differences between intact male and female controls were analyzed with an unpaired *t*-test; *p*-values < 0.05 were considered significant. Pearson correlation analyses between plasma sex hormone levels and protein expression and correlation analysis between MCT1 and CD147 expression were conducted with GraphPad Prism 10.3.1 (San Diego, CA, USA).

## 3. Results

### 3.1. Plasma Sex Hormone Concentrations

Plasma sex hormone concentrations are reported in Table 1. OVX and CST estrogen-, progesterone-, and combination-treated rats had higher plasma sex hormone concentrations than respective placebo groups. Plasma testosterone concentrations for OVX and CST testosterone rats were 6.01 ± 2.237 and 5.832 ± 1.8 ng/mL, respectively. Testosterone was undetected in OVX and CST rats treated with female sex hormones or placebo hormone pellets. Female control rats had greater plasma 17β-estradiol and progesterone levels than male controls, and undetectable testosterone levels. 17β-estradiol, progesterone, and testosterone concentrations in hormone- and placebo-treated rats, as well as intact male and female controls, were consistent with those observed following hormone treatment in age-matched post-puberty OVX and CST animals [16].

### 3.2. MCT1 Expression

MCT1 membrane expression and treatment differences are presented in [Fig pharmaceutics-17-00252-g001] and Table 2. MCT1 membrane expression was significantly different across groups (*p* = 0.0002: [Fig pharmaceutics-17-00252-g001]A, *p* = 0.0001: [Fig pharmaceutics-17-00252-g001]B, *p* = 0.0006: [Fig pharmaceutics-17-00252-g001]C, *p* = 0.0002: [Fig pharmaceutics-17-00252-g001]D, *p* = 0.0004: [Fig pharmaceutics-17-00252-g001]E, *p* = 0.0001: [Fig pharmaceutics-17-00252-g001]F, *p* = 0.0007: [Fig pharmaceutics-17-00252-g001]G, and *p* = 0.0023: [Fig pharmaceutics-17-00252-g001]H). The OVX combo trended lower compared to OVX combo placebo (*p* = 0.6262, [Fig pharmaceutics-17-00252-g001]E), while OVX/CST testosterone groups trended higher than their respective placebo groups (*p* = 0.8550: [Fig pharmaceutics-17-00252-g001]G; *p* = 0.1618: [Fig pharmaceutics-17-00252-g001]H). No significant differences in MCT1 expression were observed between the hormone-treated and hormone placebo groups. Intact females and OVX/CST groups treated with estrogen and/or progesterone, or their respective placebos, had significantly lower expression than intact males ([Fig pharmaceutics-17-00252-g001]A–F). Additionally, OVX testosterone, OVX testosterone placebo, and CST testosterone placebo groups had significantly lower expression than intact males (*p* = 0.0039 and *p* = 0.006, [Fig pharmaceutics-17-00252-g001]G; *p* = 0.0011, [Fig pharmaceutics-17-00252-g001]H).

### 3.3. CD147 Expression

CD147 membrane expression and treatment differences are presented in [Fig pharmaceutics-17-00252-g002] and Table 2. CD147 membrane expression was significantly different across OVX groups (*p* = 0.01: [Fig pharmaceutics-17-00252-g002]A, *p* = 0.05: [Fig pharmaceutics-17-00252-g002]C, and *p* < 0.0001: [Fig pharmaceutics-17-00252-g002]G). OVX estrogen-treated rats had significantly lower CD147 expression compared to OVX estrogen placebo (*p* = 0.0116, [Fig pharmaceutics-17-00252-g002]C). OVX progesterone-treated rats trended higher than OVX progesterone placebo; however, these differences were not statistically significant (*p* = 0.0735, [Fig pharmaceutics-17-00252-g002]C). OVX testosterone rats had significantly greater expression than OVX testosterone placebo (*p* = 0.0009, [Fig pharmaceutics-17-00252-g002]G). Additionally, OVX testosterone rats had significantly greater expression than both intact females (*p* < 0.0001, [Fig pharmaceutics-17-00252-g002]G) and intact males (*p* < 0.0001, [Fig pharmaceutics-17-00252-g002]G).

CD147 membrane expression was significantly different across CST groups (*p* = 0.0141: [Fig pharmaceutics-17-00252-g002]D, *p* = 0.0387: [Fig pharmaceutics-17-00252-g002]F, and *p* < 0.0001: [Fig pharmaceutics-17-00252-g002]H), with the exception of CST estrogen-treated animals (*p* = 0.0654, [Fig pharmaceutics-17-00252-g002]B). CST testosterone-treated rats had significantly greater expression than CST testosterone placebo (*p* = 0.0004, [Fig pharmaceutics-17-00252-g002]H). CST testosterone-treated rats had significantly greater expression than both intact males (*p* = 0.0001, [Fig pharmaceutics-17-00252-g002]H) and intact females (*p* = 0.0004, [Fig pharmaceutics-17-00252-g002]H). CST combo rats trended modestly lower compared to CST combo placebo (*p* = 0.1918, [Fig pharmaceutics-17-00252-g002]F); however, no significant difference in CD147 expression was observed between female sex hormone-treated rats and their respective placebo groups. CST estrogen placebo, CST progesterone placebo, and CST combo placebo groups had significantly higher expression compared to intact males (*p* = 0.041: [Fig pharmaceutics-17-00252-g002]B, *p* = 0.0210: [Fig pharmaceutics-17-00252-g002]D, and *p* = 0.0255: [Fig pharmaceutics-17-00252-g002]F).

### 3.4. SMCT1 Expression

SMCT1 membrane expression and treatment differences are presented in [Fig pharmaceutics-17-00252-g003] and Table 2. SMCT1 membrane expression was significantly different across OVX groups (*p* < 0.0001: [Fig pharmaceutics-17-00252-g003]A, *p* < 0.0001: [Fig pharmaceutics-17-00252-g003]C, *p* < 0.0001: [Fig pharmaceutics-17-00252-g003]E, and *p* < 0.0001: [Fig pharmaceutics-17-00252-g003]G). OVX testosterone-treated rats had significantly greater expression compared to OVX testosterone placebo (*p* = 0.0005, [Fig pharmaceutics-17-00252-g003]G). OVX estrogen and placebo groups had significantly lower expression than intact females (*p* < 0.0001, [Fig pharmaceutics-17-00252-g003]A). OVX progesterone and progesterone placebo groups had significantly reduced expression compared to both female (*p* < 0.0001, [Fig pharmaceutics-17-00252-g003]C) and male controls (*p* = 0.0057 and *p* = 0.0005, [Fig pharmaceutics-17-00252-g003]C). OVX combo and combo placebo groups had significantly lower expression than intact females (*p* < 0.0001, [Fig pharmaceutics-17-00252-g003]E). OVX testosterone had significantly lower expression than intact females (*p* = 0.0002, [Fig pharmaceutics-17-00252-g003]G) and OVX testosterone placebo had significantly reduced expression compared to both intact female (*p* < 0.0001, [Fig pharmaceutics-17-00252-g003]G) and male controls (*p* = 0.0075, [Fig pharmaceutics-17-00252-g003]G). Intact females had significantly greater expression, compared to intact males (*p* < 0.0001, [Fig pharmaceutics-17-00252-g003]A–G).

SMCT1 membrane expression was significantly different across CST groups (*p* < 0.0001: [Fig pharmaceutics-17-00252-g003]B, *p* < 0.0001: [Fig pharmaceutics-17-00252-g003]D, *p* < 0.0001: [Fig pharmaceutics-17-00252-g003]F, and *p* < 0.0001: [Fig pharmaceutics-17-00252-g003]H). CST testosterone-treated rats had significantly greater expression compared to CST testosterone placebo (*p* < 0.0001, [Fig pharmaceutics-17-00252-g003]H). CST testosterone had significantly lower expression than intact females (*p* < 0.0001, [Fig pharmaceutics-17-00252-g003]H) and CST testosterone placebo had significantly reduced expression compared to both intact females (*p* < 0.0001, [Fig pharmaceutics-17-00252-g003]H) and intact males (*p* = 0.0079, [Fig pharmaceutics-17-00252-g003]H). CST estrogen and CST combo rats trended modestly higher compared to their respective placebo groups (*p* = 0.1098: [Fig pharmaceutics-17-00252-g003]B, *p* = 0.0959: [Fig pharmaceutics-17-00252-g003]F). CST estrogen and respective placebo groups had significantly lower expression than intact females (*p* < 0.0001, [Fig pharmaceutics-17-00252-g003]B). CST progesterone and progesterone placebo groups had significantly reduced expression compared to both female (*p* < 0.0001, [Fig pharmaceutics-17-00252-g003]D) and male controls (*p* = 0.0005, *p* = 0.0124, [Fig pharmaceutics-17-00252-g003]D). CST combo and corresponding placebo groups had significantly lower expression than intact females (*p* < 0.0001, [Fig pharmaceutics-17-00252-g003]H).

### 3.5. Correlation Analysis

Pearson correlation analyses are presented in Appendix A. MCT1 membrane expression was positively correlated (r = 0.2416) with CD147 expression (*p* = 0.0118, Appendix A). MCT1 expression was positively correlated (r = 0.2944) with testosterone concentrations (*p* = 0.0036, Appendix A). No significant correlations were observed for female sex hormones and MCT1 expression. CD147 expression was negatively correlated (r = −0.2344) with plasma 17β-estradiol levels (*p* = 0.0215, Appendix A) and positively correlated (r = 0.6207) with testosterone levels (*p* < 0.0001, Appendix A). SMCT1 expression was positively correlated (r = 0.3785) with plasma testosterone (*p* = 0.0001, Appendix A). All evaluated target proteins had no significant correlation with progesterone levels (Appendix A).

## 4. Discussion

Differences in monocarboxylate transporter expression can alter the renal clearance of substrates. We have previously demonstrated changes in γ-hydroxybutyrate (GHB) toxicokinetics and renal clearance with testosterone treatment [23] and changes in renal MCT/SMCT expression following sex and cross-sex hormone treatment in rats that underwent gonadectomy after the onset of puberty [16]. In the present study, prepubertal ovariectomy and castration was conducted to evaluate sex and cross-sex hormone-driven regulation in the absence of initial pubertal hormone exposure. 17β-estradiol-, progesterone-, and testosterone-mediated changes in renal MCT1, CD147, and SMCT1 membrane protein expression were observed in rats with ovariectomy and castration before the onset of puberty.

Treatment with estrogen, progesterone, or testosterone did not significantly alter MCT1 expression when OVX and CST occurred before the onset of puberty ([Fig pharmaceutics-17-00252-g001]A–H); MCT1 expression trended lower in OVX rats treated with a combination of estrogen and progesterone ([Fig pharmaceutics-17-00252-g001]E). These data are consistent with post-puberty studies, in which 17β-estradiol or progesterone alone did not significantly downregulate expression, but OVX rats treated with 17β-estradiol and progesterone combined had significantly reduced expression compared to placebo [16]. However, in contrast to post-puberty gonadectomy, no significant negative correlations between MCT1 expression and plasma 17β-estradiol concentrations were observed in the present study (Appendix A). Altogether, the results suggest that downregulation of MCT1 expression in rats OVX before and after puberty requires both estrogen and progesterone, but the effects of female sex hormones may be reduced when ovariectomy occurs prepubertally. A similar diminished effect when compared to post-puberty studies was observed with testosterone treatment. MCT1 expression trended higher in OVX and CST rats treated with testosterone compared to their respective placebo controls ([Fig pharmaceutics-17-00252-g001]G,H), and MCT1 expression demonstrated a significant positive correlation with testosterone plasma concentration (Appendix A). This testosterone-mediated effect is consistent with the significantly greater MCT1 expression observed in intact males relative to intact females ([Fig pharmaceutics-17-00252-g001]A). OVX rats treated with testosterone had significantly greater expression compared to both placebo-treated and intact males [16]. The trend of testosterone-driven upregulation of MCT1 expression was observed in rats OVX before and after the onset of puberty. However, following testosterone treatment, rats OVX before puberty had a reduced percent change in MCT1 expression compared to rats OVX post puberty. The decreased change in expression suggests the absence of initial exposure to pubertal hormones diminishes the effect of testosterone on MCT1 upregulation in OVX rats.

CD147 functions as the ancillary protein for MCT1, and the association of the two proteins enhances MCT1 membrane expression [1,5]. Testosterone treatment altered CD147 expression in rats that underwent OVX and CST prior to puberty ([Fig pharmaceutics-17-00252-g002]G,H), while estrogen treatment altered expression in OVX alone ([Fig pharmaceutics-17-00252-g002]A). CD147 expression displayed a significant positive correlation with plasma testosterone concentrations (Appendix A) and a significant negative correlation with plasma 17β-estradiol levels (Appendix A), consistent with the observed trends in protein expression. CD147 membrane expression in OVX rats treated with estrogen was significantly reduced compared to placebo controls ([Fig pharmaceutics-17-00252-g002]A), suggesting estrogen is involved in the downregulation of CD147 in OVX rats; however, this difference in CD147 expression was not exhibited in rats that underwent OVX post-puberty [16]. This suggests a more pronounced effect of estrogen treatment alone in rats that have not gone through puberty. Additionally, both OVX and CST rats treated with testosterone had significantly greater expression than the respective placebo groups ([Fig pharmaceutics-17-00252-g002]G,H), demonstrating that testosterone mediated upregulation of CD147 expression following both sex- and cross-sex treatment. When OVX occurred after puberty, testosterone treatment resulted in a significant increase in expression, compared to the OVX placebo group [16], consistent with prepuberty results. In contrast, no significant difference was observed, when CST occurred post-puberty, between CST rats treated with testosterone and those treated with placebo [16], indicating differential effects of testosterone in rats CST pre- and post-puberty.

SMCT1 expression was significantly upregulated in response to sex- and cross-sex testosterone treatment in rats with OVX and CST before puberty ([Fig pharmaceutics-17-00252-g003]G,H) and demonstrated a significant positive correlation with plasma testosterone concentrations (Appendix A). Testosterone treatment had a differential effect when OVX and CST occurred after puberty, with significantly downregulated expression in CST rats treated with testosterone compared placebo-treated animals, while OVX rats had modestly higher expression versus placebo [16]. Taken together, the results suggest testosterone has differential effects when CST occurs before or after puberty, with a significant upregulation observed in prepuberty studies and significant downregulation observed in post-puberty evaluations. Hosoyamada et al. investigated the contribution of androgens to hyperuricemia; smct1 was evaluated due to its role in urate reabsorption [14]. Testosterone upregulated kidney smct1 expression in orchiectomized mice [14], consistent with the presently observed effects of testosterone. SMCT1 membrane expression was not significantly different between CST rats treated with female sex hormones and those treated with the respective placebo controls ([Fig pharmaceutics-17-00252-g003]A–F), and no significant positive correlations were observed with 17β-estradiol or progesterone plasma concentrations (Appendix A), in contrast to the post-puberty studies. When CST occurred after puberty, estrogen treatment significantly increased SMCT1 expression compared to placebo, and treatment with estrogen and progesterone combined significantly increased expression as compared to placebo [16]. SMCT1 expression is upregulated following cross-sex female hormone treatment when gonadectomy occurs after puberty, but is unaltered when gonadectomy occurs before puberty, demonstrating the impact of puberty on estrogen/progesterone-mediated regulation of SMCT1.

Testosterone treatment altered the toxicokinetics of GHB (an MCT/SMCT substrate) in rats that underwent OVX and CST surgery after puberty [23]. Following i.v. administration of 1500 mg/kg GHB, plasma exposure (AUC) was significantly increased in CST rats treated with testosterone, compared to placebo [23]. Additionally, the fraction of drug eliminated in urine (fe) was significantly decreased in the OVX and CST rats treated with testosterone, compared to the respective placebo controls [23]. These observed differences are consistent with the testosterone-mediated upregulation of renal MCT1 expression seen in rats CST or OVX after puberty. Future toxicokinetic studies should be conducted in rats that have undergone gonadectomy before puberty to determine the implications of the observed differences in MCT and SMCT regulation on GHB toxicokinetics and toxicity.

Future studies should evaluate the underlying mechanisms contributing to the observed differences in expression when gonadectomy is performed before or after puberty. Epigenetic regulation of MCT/SMCT expression has been evaluated in the scope of disease states [1], and represents a potential mechanism for the expression differences observed in the present study. Epigenetic regulation via DNA methylation of CpG sites has been demonstrated to influence MCT gene expression [24]. MCT1 gene expression was downregulated and upregulated in the breast cancer cell line MDA-MB-231 with CpG island methylation and 5-aza-2’-deoxycytidine-induced demethylation, respectively [24]. The effects of puberty and exogenous hormone administration on global epigenetic regulation have been investigated [17,25], but to our knowledge, there are no studies evaluating the effect of sex hormones on DNA methylation of CpG sites associated with MCT or SMCT gene expression. Han et al. demonstrated differences pre- and post-adolescence in the methylation of 15,532 CpG sites [17]. Additionally, female participants who were on oral contraceptives had an overall reduced global DNA methylation compared to participants who did not utilize contraceptives [25]. Future studies will assess DNA methylation and sex hormone regulation of MCT1/SMCT1 in human cells to evaluate the translatability of these regulatory mechanisms to the human population.

In summary, sex and cross-sex hormone-induced differences in MCT1, CD147, and SMCT1 expression were observed when gonadectomy was conducted before puberty. The presence or absence of expression differences, as well as the directionality of change relative to expression data, in animals that had undergone gonadectomy after puberty differed based on target protein and sex hormone treatment. MCT1 analysis demonstrated a reduced or absent effect of sex hormones on expression, whereas CD147 results demonstrated a retained effect of testosterone upregulation of expression. SMCT1 results showed a differential effect of testosterone on membrane-bound expression, with significant upregulation when gonadectomy occurred before puberty, and with downregulation observed with post-puberty gonadectomy. Additional investigations are necessary to understand the mechanisms responsible for these observed differences when gonadectomy occurs before or after puberty, and the consequences of MCT/SMCT expression differences on the renal clearance of drug substrates.

## Figures and Tables

**Figure 1 pharmaceutics-17-00252-g001:**
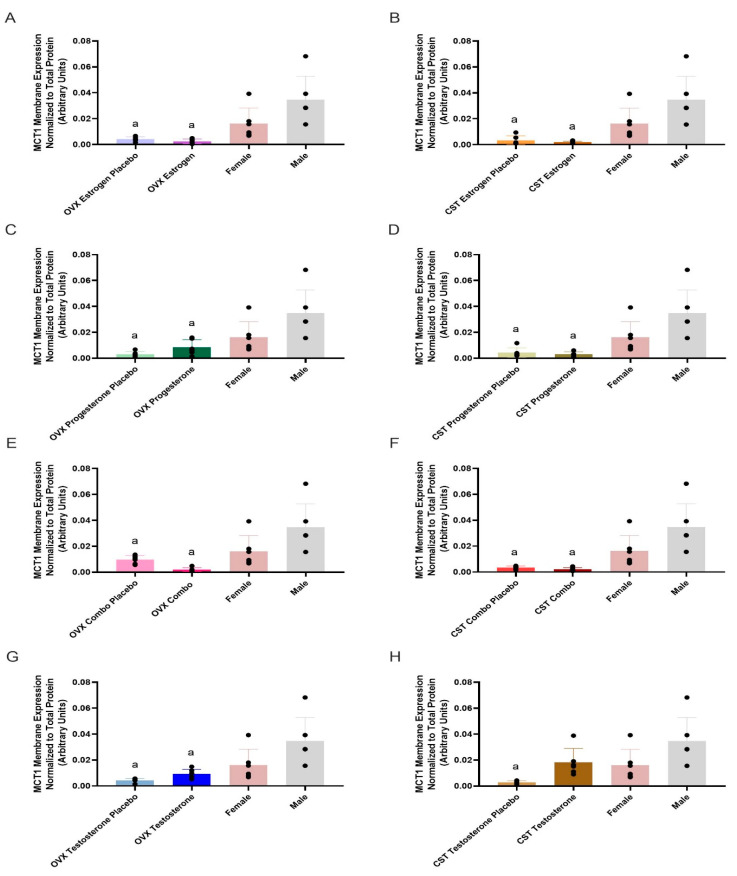
Kidney MCT1 membrane expression in OVX and CST Sprague–Dawley rats administered sex hormones or their respective placebo controls. Results compared to male and female controls. OVX (**A**) and CST (**B**) rats administered 17β-estradiol; OVX (**C**) and CST (**D**) rats administered progesterone; OVX (**E**) and CST (**F**) rats administered an estrogen and progesterone combination; OVX (**G**) and CST (**H**) rats administered testosterone. Results presented as mean ± SD with individual data points, N = 6. ^a^ *p* < 0.05 compared to Male.

**Figure 2 pharmaceutics-17-00252-g002:**
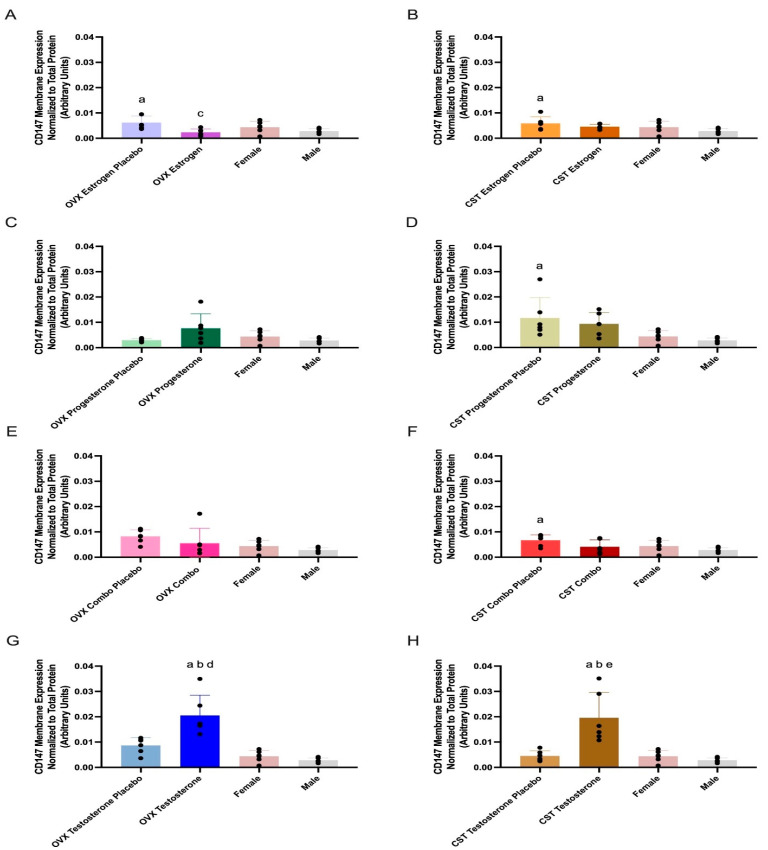
Kidney CD147 membrane expression in OVX and CST Sprague–Dawley rats administered sex hormones or respective placebo controls. Results compared to male and female controls. OVX (**A**) and CST (**B**) rats administered 17β-estradiol; OVX (**C**) and CST (**D**) rats administered progesterone; OVX (**E**) and CST (**F**) rats administered an estrogen and progesterone combination; OVX (**G**) and CST (**H**) rats administered testosterone. Results presented as mean ± SD with individual data points, N = 6. ^a^ *p* < 0.05 compared to Male. ^b^ *p* < 0.05 compared to Female. ^c^ *p* < 0.05 compared to OVX Estrogen Placebo. ^d^ *p* < 0.05 compared to OVX Testosterone Placebo. ^e^ *p* < 0.05 compared to CST Testosterone Placebo.

**Figure 3 pharmaceutics-17-00252-g003:**
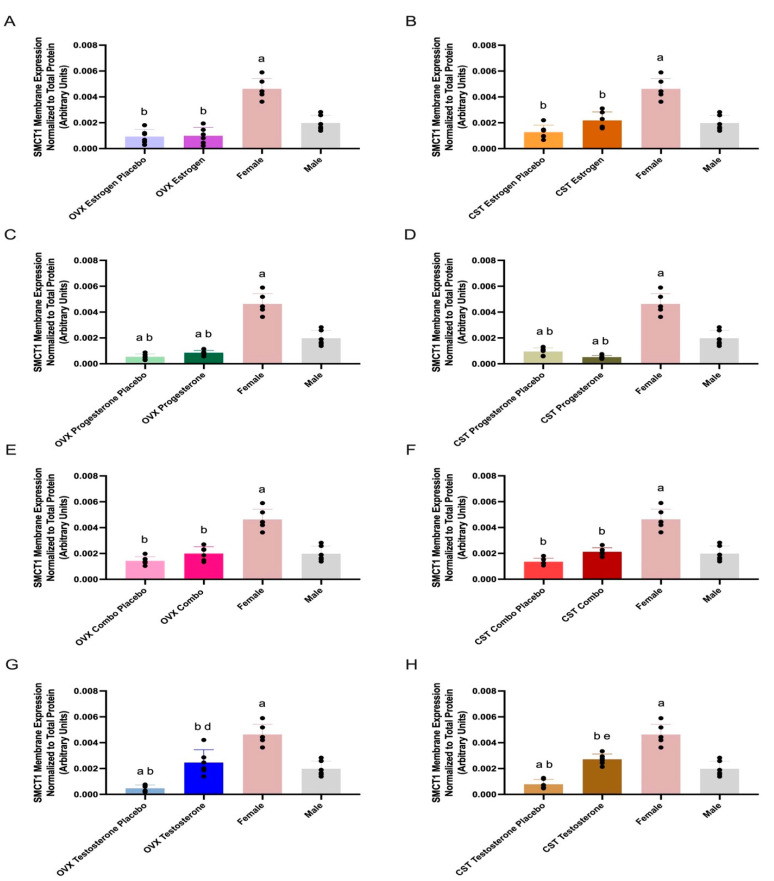
Kidney SMCT1 membrane expression in OVX and CST Sprague–Dawley rats administered sex hormones or respective placebo controls. Results compared to male and female controls. OVX (**A**) and CST (**B**) rats administered 17β-estradiol; OVX (**C**) and CST (**D**) rats administered progesterone; OVX (**E**) and CST (**F**) rats administered estrogen and progesterone combination; OVX (**G**) and CST (**H**) rats administered testosterone. Results presented as mean ± SD with individual data points, N = 6. ^a^ *p* < 0.05 compared to Male. ^b^ *p* < 0.05 compared to Female. ^d^ *p* < 0.05 compared to OVX Testosterone Placebo. ^e^ *p* < 0.05 compared to CST Testosterone Placebo.

**Table 1 pharmaceutics-17-00252-t001:** Plasma 17β-Estradiol, progesterone, and testosterone concentrations in hormone- or placebo-treated OVX and CST rats, and female and male controls. Data presented as mean ± SD, N = 6.

Plasma Hormone Concentration	17β-Estradiol (pg/mL)	17β-Estradiol Non-Detected	Progesterone (ng/mL)	Progesterone Non-Detected	Testosterone (ng/mL)	Testosterone Non-Detected
OVX Estrogen Placebo	8.94 ± 6.495	0 out of 6	3.813 ± 3.605	0 out of 6	0	6 out of 6
OVX Estrogen	1199 ± 490.1	0 out of 6	4.368 ± 4.676	0 out of 6	0	6 out of 6
OVX Progesterone Placebo	7.769 ± 2.163	0 out of 6	2.106 ± 1.762	0 out of 6	0	6 out of 6
OVX Progesterone	9.988 ± 7.323	0 out of 6	7.596 ± 5.313	0 out of 6	0	6 out of 6
OVX Combo Placebo	14.28 ± 9.546	0 out of 6	1.969 ± 1.154	0 out of 6	0	6 out of 6
OVX Combo	1023 ± 552.2	0 out of 6	11.36 ± 4.661	0 out of 6	0	6 out of 6
OVX Testosterone Placebo	8.187 ± 4.758	0 out of 6	2.201 ± 1.376	0 out of 6	0	6 out of 6
OVX Testosterone	9.155 ± 1.556	0 out of 6	1.033 ± 0.5029	0 out of 6	6.01 ± 2.237	0 out of 6
Female (Estrus)	20.97 ± 14.65	0 out of 6	8.026 ± 2.4	0 out of 6	0	6 out of 6
CST Estrogen Placebo	5.899 ± 4.138	0 out of 6	1.363 ± 0.5882	0 out of 6	0	6 out of 6
CST Estrogen	1159 ± 414.3	0 out of 6	2.839 ± 2.553	0 out of 6	0	6 out of 6
CST Progesterone Placebo	13.51 ± 4.177	0 out of 6	1.296 ± 0.7488	0 out of 6	0	6 out of 6
CST Progesterone	10.33 ± 4.506	0 out of 6	6.661 ± 4.244	0 out of 6	0	6 out of 6
CST Combo Placebo	6.126 ± 4.356	0 out of 6	1.481 ± 0.8062	1 out of 6	0	6 out of 6
CST Combo	1447 ± 534.9	0 out of 6	8.665 ± 4.866	0 out of 6	0	6 out of 6
CST Testosterone Placebo	6.522 ± 2.987	1 out of 6	1.414 ± 0.948	1 out of 6	0	6 out of 6
CST Testosterone	33.52 ± 12.38	0 out of 6	1.899 ± 1.442	0 out of 6	5.832 ± 1.8	0 out of 6
Male	17.2 ± 5.654	0 out of 6	1.791 ± 1.832	3 out of 6	2.099 ± 0.4514	1 out of 6

**Table 2 pharmaceutics-17-00252-t002:** Percentage change in kidney MCT1, CD147, and SMCT1 expression following sex and cross-sex hormone treatment in OVX and CST rats, and in females relative to males. N = 6 per treatment group.

	Treatment vs. Placebo	MCT1	CD147	SMCT1
OVX	17β-estradiol	−42.6	−62	6.5
	Progesterone	177.3	159.7	56.9
	17β-estradiol/Progesterone	−79.7	−32.5	39.4
	Testosterone	119.7	137.3	423.8
CST	17β-estradiol	−43.2	−22.2	70.9
	Progesterone	−29.9	−20	−45.5
	17β-estradiol/Progesterone	−39.9	−38.5	56.4
	Testosterone	524.7	332.5	246.4
Female vs. Male		−53.4	55.4	133.4

## Data Availability

The data presented in this study are available within the article and Appendix A.

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
