# Peer review of "Effects of Sex and Cross-Sex Hormone Treatment on Renal MCT/SMCT Expression Following Prepubertal Gonadectomy"

_pharmaceutics, 2025, doi:10.3390/pharmaceutics17020252_

Round 1
Reviewer 1 Report
Comments and Suggestions for Authors
The authors’ manuscript focuses on the evaluation of Prepubertal gonadectomy on renal proton- and sodium-dependent monocarboxylate transporters expression. Protein expression of MCT1, CD147, and SMCT1 was analyzed and authors found that their expression is regulated by sex hormones.
Major issues:
-
The relevance of the study or the potential translation of the obtained results is not clear. What is the main issue that the authors expect to respond to?
-
Previous work of the authors have already demonstrated the regulation of MCT/SMCT expression by sex hormones. Similar results were obtained in the present study. The novelty of this study is that the analysis was performed before puberty. What is the relevance of carrying on this analysis before puberty? How will this contribute to better understanding of drug pharmacokinetics?
-
Figures 1, 2 and 3 present different graphics showing the effects of gonadectomy and sex hormones treatments on the protein expression of MCT1, CD147, and SMCT1. Please consider to merge the results regarding OVX and treatments in just a graphic (for each protein) and the same to CST samples.
-
In Discussion topic, the results description is repeated and extensive. On the other hand, epigenetic regulation is introduced at the Discussion for the first time in the manuscript and their relevance to the results obtained is not clear.
Minor:
-
Figure 1. A-H, Female and male data are the same, but in H) is missing supposed statistical significance between Female and Male.
Author Response
We thank the reviewers for their comments on our manuscript. We have addressed the concerns below, and made changes to the manuscript (highlighted in yellow).
Major issues:
- The relevance of the study or the potential translation of the obtained results is not clear. What is the main issue that the authors expect to respond to?
- One of the aims of this study was to evaluate the impact of puberty on MCT/SMCT regulation in response to sex hormones. Previous studies have demonstrated that epigenetic mechanisms of gene regulation, specifically DNA methylation, are altered during and after puberty. MCT1 and SMCT1 have been shown to be regulated by DNA methylation in humans and rats, so our hypothesis was that we may see a differential response to sex hormones in animals that have not gone through puberty. The introduction has been updated to more clearly reflect this.
- Further studies are needed to look at translation of these potential regulatory mechanisms to humans. A statement specifically addressing future studies has been added to the discussion section
- Previous work of the authors have already demonstrated the regulation of MCT/SMCT expression by sex hormones. Similar results were obtained in the present study. The novelty of this study is that the analysis was performed before puberty. What is the relevance of carrying on this analysis before puberty? How will this contribute to better understanding of drug pharmacokinetics?
- The present study demonstrated differential regulation of MCT/SMCT transporters in response to sex and cross-sex hormone treatment when gonadectomy was conducted prior to puberty. This suggests that individuals that do not experience puberty may have altered GHB toxicokinetics. A number of situations may lead to an individual not going through puberty including the administration of hormone blockers or conditions leading to altered hormone production (for example Klinefelter syndrome). Additionally, there is the potential for altered pharmacokinetics of other compounds that are MCT/SMCT substrates.
- Figures 1, 2 and 3 present different graphics showing the effects of gonadectomy and sex hormones treatments on the protein expression of MCT1, CD147, and SMCT1. Please consider to merge the results regarding OVX and treatments in just a graphic (for each protein) and the same to CST samples.
- The graphical display of transporter expression results is consistent with our previous kidney publication to allow comparison between the two studies. The results are separated by hormone treatment, as we were not directly comparing the individual hormone treatments.
- In Discussion topic, the results description is repeated and extensive. On the other hand, epigenetic regulation is introduced at the Discussion for the first time in the manuscript and their relevance to the results obtained is not clear.
- Information regarding changes in epigenetic regulation during puberty was added to the introduction section (see comments above).
Minor:
- Figure 1. A-H, Female and male data are the same, but in H) is missing supposed statistical significance between Female and Male.
-
- We reviewed the statistical analysis for figure 1. Intact males and females were significantly different based on post-hoc analysis with the exception of the data presented in Figure 1H. The intact control data was re-evaluated using an unpaired t-test to evaluate for differences between males and females, which resulted in a p-value of 0.06. Figure 1 was updated to remove the significant difference notation between males and females.
Reviewer 2 Report
Comments and Suggestions for Authors
In the manuscript entitled “Effects of Sex and Cross-sex Hormone Treatment on Renal MCT/SMCT Expression Following Prepubertal Gonadectomy” the authors investigate the effects of sex and cross-sex hormone treatments on the renal expression of monocarboxylate transporters (MCTs) and sodium-dependent monocarboxylate transporters (SMCTs) in prepubertal gonadectomized rats. It provides novel insights into the role of pubertal hormone exposure in the regulation of renal MCT1, CD147, and SMCT1, highlighting the pharmacokinetic implications of these findings.
The study uses a robust animal model with prepubertal gonadectomy and detailed hormone treatment protocols.
1. How many animals were used in each experimental group? The authors should specify those information’s?
2. How did you determine the concentration of the 17β-estradiol, progesterone, and testosterone?
3. The authors should add appropriate western blot images to the Figure 1 and Figure 2.
4. The results are very confusingly written. The authors determined protein expressions of CD147 and MCT1. Proteins were isolated from the membrane.
Comments on the Quality of English Language
Ok
Author Response
We thank the reviewers for their comments on our manuscript. We have addressed the concerns below, and made changes to the manuscript (highlighted in yellow).
- How many animals were used in each experimental group? The authors should specify those information’s?
- Six animals were used per treatment group. A statement reflecting this was added to the methods section (subsection 2.2, page 7), and the group size is indicated in all figure legends and tables.
- How did you determine the concentration of the 17β-estradiol, progesterone, and testosterone?
- 17B-estradiol concentrations were determined by a commercial ELISA kit. Progesterone and testosterone concentrations were determined by a validated LCMSMS assay. The assays are described in Subsection 2.3 on page 8.
- The authors should add appropriate western blot images to the Figure 1 and Figure 2.
- All western blot images were uploaded as a supplementary files, including total protein images which were used for data normalization.
- The results are very confusingly written. The authors determined protein expressions of CD147 and MCT1. Proteins were isolated from the membrane
- We have added a table (Table 2) to the results section summarizing the percentage differences between treatment groups. We have removed the corresponding data from the manuscript to improve the readability of the results.